# Relationship between Being Overweight and Clinical Outcomes of Ablation Therapy for Hepatocellular Carcinoma under Ultrasound Guidance: A Retrospective Analysis

**DOI:** 10.3390/cancers15041289

**Published:** 2023-02-17

**Authors:** Takeshi Hatanaka, Yutaka Yata, Naoto Saito, Sachi Nakano, Yuya Nakano, Yoichi Hazama, Sachiko Yoshida, Yoko Hachisu, Yoshiki Tanaka, Teruo Yoshinaga, Atsushi Naganuma, Satoru Kakizaki

**Affiliations:** 1Department of Gastroenterology, Gunma Saiseikai Maebashi Hospital, Maebashi 371-0821, Japan; 2Department of Gastroenterology, Hanwa Memorial Hospital, Osaka 558-0041, Japan; 3Department of Gastroenterology, National Hospital Organization Takasaki General Medical Center, Takasaki 370-0829, Japan; 4Department of Clinical Research, National Hospital Organization Takasaki General Medical Center, Takasaki 370-0829, Japan; 5Department of Gastroenterology and Hepatology, Gunma University Graduate School of Medicine, Maebashi 371-8511, Japan

**Keywords:** overweight, hepatocellular carcinoma, radiofrequency ablation, microwave ablation, body mass index, technical success rate, obesity paradox

## Abstract

**Simple Summary:**

In this retrospective study of 198 HCC patients receiving ablation therapy, participants were divided into two groups based on body mass index (BMI): overweight patients (BMI ≥ 25) and non-overweight patients (BMI < 25). The technical success rate (TSR) in the first session was significantly higher in non-overweight patients than in overweight patients (90.3% vs. 78.4%, *p* = 0.03). Fifteen (20.3%) overweight and eleven (8.9%) non-overweight patients required additional ablation therapy for residual tumors, resulting in almost similar TSRs at the final session (95.9% vs. 99.2%, *p* = 0.3). Local tumor progression and distant recurrence rates were not significantly different between the two groups, but overall survival was better in overweight patients than in non-overweight patients (*p* < 0.001). Despite the potential adverse impact of being overweight on public health problems, the present findings showed the relationship between being overweight and improved survival. The negative aspects of being overweight might remain as minor technical issues in HCC patients receiving ablation therapy.

**Abstract:**

This study aimed to investigate the effect of being overweight on the outcome of ablation therapy for patients with early-stage hepatocellular carcinoma (HCC). This retrospective study included 198 patients with HCC who underwent radiofrequency ablation or microwave ablation at Gunma Saiseikai Maebashi Hospital between April 2017 and December 2021. We divided the patients into two groups based on their body mass index (BMI): overweight (BMI ≥ 25 kg/m^2^, *n* = 74 (37.4%)) and non-overweight (BMI < 25 kg/m^2^, *n* = 124 (62.6%)). The technical success rates (TSRs) in the first session were 78.4% and 90.3% in overweight and non-overweight patients, respectively, with a significant difference (*p* = 0.03). Additional ablation therapy for residual tumors was required in 15 (20.3%) overweight and 11 (8.9%) non-overweight patients (*p* = 0.03), resulting in 95.9% and 99.2% TSRs at the final session, respectively, without a significant difference (*p* = 0.3). While local tumor progression and distant recurrence rates were not significantly different between the two groups, overall survival was better in overweight patients than in non-overweight patients (*p* < 0.001). Despite the potential adverse impact of being overweight on public health problems, the present findings showed the relationship between being overweight and improved survival. The negative aspects of being overweight might remain as minor technical issues in HCC patients receiving ablation therapy.

## 1. Introduction

Liver cancer is the second most frequent cause of cancer-related death worldwide [1] and hepatocellular carcinoma (HCC) is the most common type of liver cancer. According to the Barcelona Clinic Liver Cancer staging system [2] and the Japan Society of Hepatology guidelines [3], ablation therapy is indicated for Child–Pugh class A or B patients with a tumor size ≤3 cm in diameter and ≤3 nodules. Radiofrequency ablation (RFA) shows good local antitumor effects and a safe profile, resulting in long-term survival [4,5,6]. According to a recent randomized controlled trial comparing the effectiveness of RFA and surgical resection in patients with early-stage HCC, recurrence-free survival was not significantly different between the two treatment modalities [7].

The prevalence of being overweight/obesity is increasing [8], and given its impact on an increased risk of metabolic comorbidities, cardiovascular diseases [9], and non-alcoholic fatty liver disease [10], it continues to gain more attention in disease management. In Japan, the focus is shifting to the impact of being overweight/obesity on HCC, along with the proportion of non-viral chronic liver disease [11,12]. Many physicians in Japan have conducted ablation therapy under real-time ultrasonographic guidance. However, subcutaneous fat reduces the resolution of ultrasound imaging, resulting in poor clinical outcomes. This study aimed to investigate the effect of being overweight on the outcomes of ablation therapy for patients with early-stage HCC.

## 2. Materials and Methods

### 2.1. Patients

From April 2017 to December 2021, 219 patients with hepatic malignant tumors were treated with RFA or microwave ablation (MWA) at the Gunma Saiseikai Maebashi Hospital. We excluded 21 patients with metastatic tumors, and the remaining 198 patients with HCC were enrolled into the study (Figure 1). The diagnosis of HCC was confirmed by pathological or typical findings on radiological imaging, including contrast-enhanced dynamic computed tomography (CT) and gadolinium ethoxybenzyl diethylenetriamine pentaacetic acid (Gd-EOB-DTPA)-enhanced magnetic resonance imaging (MRI) [13,14]. None of the patients had ascites, which might have affected their body mass index (BMI).

### 2.2. Indication for Ablation Therapy and Ablation Procedure

In general, our indications for ablation therapy were as follows: (a) tumor within three or fewer nodules and less than 3 cm in diameter, (b) no radiological evidence of tumor involvement in the major portal and hepatic veins, (c) no evidence of extrahepatic metastases, and (d) Child–Pugh class A or B disease. However, for some patients who did not meet these conditions, we conducted ablation therapy if they were likely to benefit from prolonged survival.

Two hepatologists with 10 and 5 years of experience conducted RFA and MWA procedures via real-time ultrasound (Aplio500, Canon Medical Systems, Ohtawara, Japan) using a convex transducer ultrasound probe. We intravenously injected 15 mg of pentazocine and 3 mg of midazolam before the procedure. When patients were inadequately sedated, we infused midazolam with careful observation until appropriate sedation was obtained. We also intravenously administered 15 mg pentazocine when patients seemed to experience pain during the procedure. We performed artificial pleural effusion and/or the artificial ascites technique in patients with tumors below the hepatic dome and adjacent to other organs, including the gastrointestinal tract. In general, we used a fusion imaging technique to identify the precise tumor location. We also used contrast-enhanced ultrasonography (CEUS) when the tumors were not clearly visible, despite fusion imaging. For the MWA, we used a 13-gauge, internally saline-cooled coaxial antenna (Emprint^TM^ System; Covidien, Boulder, CO, USA). The antenna was inserted into a targeted tumor, and the output energy was applied. We started at 45 W for 1 min and then gradually increased to 60 W for 1 min, 75 W for 1 min, and 100 W for 3.5–8.5 min to obtain optimal necrosis. For RFA, we used a 17-gauge internally cooled length-adjustable electrode (VIVA RF System; STARmed, Gyeonggi-do, Goyang-si, Republic of Korea). The active length of the tip was determined based on the size of the target tumor. The electrode was inserted into the targeted tumor, and radiofrequency energy was delivered. We started with 40 W for the 2 cm exposed tip and 60 W for the 3 cm exposed tip, and the output energy was gradually increased at a rate of 20 W per minute.

We performed enhanced dynamic CT the day after ablation therapy. When a residual portion of the targeted tumor was suspected, we performed additional ablation therapy.

### 2.3. The Definition of Being Overweight

We measured the patients’ height and body weight on hospital admission the day before the ablation therapy. BMI was calculated by dividing the baseline weight (kg) by the square of the height (m^2^). We defined patients with BMI ≥ 25 kg/m^2^ as overweight patients, and those with BMI < 25 kg/m^2^ as non-overweight based on the criteria of the World Health Organization [15].

### 2.4. Follow-Up and Evaluation of Efficacy and Safety of Ablation Therapy

We measured the serum levels of α-fetoprotein (AFP), des-gamma-carboxy prothrombin (DCP), and lens culinaris agglutinin–reactive fraction of α-fetoprotein (AFP-L3) every 1–3 months and performed dynamic CT or EOB-MRI every 3 months to detect HCC recurrence. The technical success rate (TSR) was defined as the proportion of patients who achieved complete ablation of the targeted nodules, which was assessed using enhanced dynamic CT the day after ablation therapy. Local tumor progression was defined as the appearance of an HCC tumor adjacent to the initially treated tumors and distant recurrence as the development of HCC nodules away from the treated site. Overall survival (OS) was defined as the period from the day of ablation therapy to death from any cause. Major complications were assessed according to a previous report [16].

### 2.5. Statistical Analyses

All statistical analyses were conducted using EZR software program version 1.61 (Saitama Medical Center, Jichi Medical University, Saitama, Japan) [17]. Categorical and numerical variables are reported as numbers (percentages) and medians (interquartile ranges (IQR)). Comparative analyses between overweight and non-overweight patients were performed using the chi-squared test, Fisher’s exact test, or Mann–Whitney U test. Kaplan–Meier curves were generated and compared using the log-rank test. Cox proportional regression was used to assess the hazard ratio (HR) and *p*-value. We adopted the inverse probability of the treatment weight (IPTW) to adjust for imbalances in the baseline characteristics of overweight and non-overweight patients. We created a propensity score for BMI ≥ 25 kg/m^2^ using a logistic regression model. The following variables were used: age, male sex, viral-related disease, performance status (0 vs. 1 or 2), treatment naïve, total bilirubin, serum albumin, Child–Pugh classification (A vs. B or C), albumin-bilirubin grade (ALBI grade) (1 vs. 2 or 3), maximum tumor diameter (≥20 mm vs. <20 mm), number of tumors (solitary vs. multiple), AFP (≥100 ng/mL vs. <100 ng/mL), AFP-L3 (≥20% vs. <20%), DCP (≥100 mAU/mL vs. <100 mAU/mL), and tumor location (left or right lobe vs. both lobes). Viral-related diseases were defined as hepatitis B virus (HBV) or hepatitis C virus (HCV) infections. We excluded patients in the bottom or top 1% percentile of the propensity score to avoid extreme weights. Inverse probability weights were calculated as the proportion of overweight patients/propensity score for overweight patients and as (1 − the proportion of overweight patients)/(1 − propensity score) for non-overweight patients. The cross-sectional areas of subcutaneous adipose tissue (SAT) and visceral adipose tissue (VAT) were measured based on pretreatment CT at the L3 inferior endplate level using the SYNAPSE VINCENT software program (Fuji Film Medical Co., Ltd., Tokyo, Japan). The subcutaneous adipose tissue index (SATI) and visceral adipose tissue index (VATI) were calculated by dividing the VAT and SAT by the square of height (m^2^). Statistical significance was set at *p* < 0.05.

## 3. Results

### 3.1. Patient Characteristics

The median age of the patients was 74.0 (IQR 69.0–79.0) years, with 142 (71.7%) male patients. The median BMI was 23.4 (22.0–26.5) kg/m^2^. The underlying liver diseases were HBV, HCV, alcohol, and others in 11 (5.6%), 110 (55.6%), 42 (21.2%), and 35 (17.7%) patients, respectively, resulting in 121 (61.1%) cases of viral-related diseases. Approximately 80% of the patients were determined to have an ECOG Performance Status of 0. The Child–Pugh class was A, B, and C in 174 (87.9%), 22 (11.1%), and 2 (1.0%) patients, respectively. The median ALBI score was −2.57 (−2.84 to −2.23), with half of the patients being ALBI grade 1 and 2, respectively. The maximum tumor diameter was 18 (12–25) mm, and approximately 60% of the patients had solitary nodules. RFA and MWA were performed in 138 (69.7%) and 60 (30.3%) patients, respectively. We divided the patients into two groups based on baseline BMI: overweight patients (BMI ≥ 25 kg/m^2^, *n* = 74 (37.4%)) and non-overweight patients (BMI < 25 kg/m^2^, *n* = 124 (62.6%)). Among the non-overweight patients, 21 (10.6%) patients were underweight (BMI < 20 kg/m^2^) and 103 (52.2%) patients were normal-weight (20 < BMI < 25 kg/m^2^).

No significant differences were observed between overweight and non-overweight patients, except for underlying liver disease (*p* = 0.02) and viral-related disease (*p* = 0.04; Table 1).

### 3.2. Short-Term Clinical Outcomes and Factors Affecting TSR at the First Session

The TSRs in the first session were 78.4% and they were 90.3% in the overweight and non-overweight patients, respectively. The TSR in the first session was significantly lower in overweight patients than in non-overweight patients (*p* = 0.03). Additional ablation therapy for residual tumors was required in 15 (20.3%) overweight and 11 (8.9%) non-overweight patients (*p* = 0.03), resulting in 95.9% and 99.2% TSRs at the final session, respectively. The difference in the TSR at the final session was comparable between the two groups (*p* = 0.3). The frequency of CEUS use was significantly lower in overweight patients than in non-overweight patients (*p* = 0.02). The number of insertion and total ablation time per case were not significantly different between the two groups (*p* = 0.3 and 0.5, respectively). Major complications developed in one (1.4%) overweight and two (1.6%) non-overweight patients (*p* = 1.0; Table 2).

Next, we performed subgroup analyses of tumor factors affecting the TSR in the first session (Table 3). Among patients with tumors ≥ 20 mm in diameter, the TSR at the first session was numerically lower in overweight patients than in non-overweight patients, without statistical significance (64.5% vs. 83.9%, *p* = 0.06). Regarding tumors spreading to both lobes, the TSR in the first session was significantly lower in overweight patients than in non-overweight patients (31.2% vs. 75.0%, *p* = 0.03), but no significant group difference was observed for HCC limited to the left or right lobe (left lobe, 81.2% vs. 90.0%, *p* = 0.6; right lobe, 95.1% vs. 93.2%, *p* = 1.0). Despite no significant difference in the TSR between overweight and non-overweight patients with solitary tumors (93.0% vs. 95.0%, *p* = 0.7), it was significantly lower in overweight patients with multiple (≥2) nodules than in the corresponding non-overweight patients (58.1% vs. 81.8%, *p* = 0.04). Among patients with multiple nodules that spread to both lobes, the TSR at the first session was significantly lower in overweight patients than in non-overweight patients (31.2% vs. 75.0%, *p* = 0.03), whereas it was not significantly different between overweight and non-overweight patients with multiple nodules that were limited to the left or right lobe (86.7% vs. 85.7%, *p* = 1.0).

We assessed the values of SATI and VATI based on pretreatment CT in 139 male patients and calculated the diagnostic performance of the TSR in the first session. SATI had an area under the receiver operating characteristic curve (AUROC) of 0.64 (95% confidence interval (CI) 0.52–0.76), with a specificity of 56.0% and sensitivity of 73.9% (Figure 2a), whereas VATI showed an AUROC of 0.61 (95% CI 0.48–0.74), with a specificity of 56.0% and a sensitivity of 73.9% (Figure 2b). BMI had an AUROC of 0.62 (95% CI 0.48–0.76), with a specificity of 71.6% and a sensitivity of 56.5% (Figure 2c). Although the AUROC of SATI was numerically higher than that of VATI and BMI, statistical significance was not observed among the three groups (SATI vs. VATI, *p* = 0.6; SATI vs. BMI, *p* = 0.6; VATI vs. BMI, *p* = 0.9).

### 3.3. Local Tumor Progression, Distant Recurrence, and Overall Survival

We excluded three overweight and one non-overweight patient from the analysis of local tumor progression, distant recurrence, and OS because of incomplete ablation in these patients. In the analysis of local tumor progression, 45 events (22.7%) occurred during the observation period. In the crude cohort, the 1-, 2-, 3-, and 4-year local tumor progression rates were 18.3% (95% CI 12.2–27.0), 26.5% (95% CI 18.9–36.5), 30.3% (95% CI 21.8–41.4), and 31.4% (95% CI 47.5–23.9) in the non-overweight patients and 13.9% (95% CI 7.5–25.0), 18.2% (95% CI 10.4–30.8), 20.5% (95% CI 12.0–33.6), and 32.7% (95% CI 19.8–51.0) in the overweight patients, respectively. The difference between the two groups was not statistically significant (HR 0.70, 95% CI 0.38–1.31, *p* = 0.2; Figure 3a). In the weighted cohort, the 1-, 2-, 3-, and 4-year local tumor progression rates were 15.6% (95% CI 9.7–24.7), 24.9% (95% CI 17.0–35.6), 27.4 (95% CI 18.7–39.2), and 27.4 (95% CI 18.7–39.2) in the non-overweight patients and 13.0% (95% CI 6.6–24.7), 17.7% (95% CI 9.7–31.2), 20.1 (95% CI 11.3–34.3), and 34.2 (95% CI 19.7–55.1) in the overweight patients, respectively. The difference between the two groups was not statistically significant (HR 0.90, 95% CI 0.69–1.20, *p* = 0.50; Figure 3b).

Distant recurrence was detected in 98 (49.9%) patients. The 1-, 2-, 3-, and 4-year distant recurrence rates were 37.1% (95% CI 28.8–47.0), 53.8% (95% CI 43.9–64.4), 60.7% (95% CI 49.4–72.0), and 64.2% (95% CI 52.0–76.3) in the non-overweight patients and 38.6% (95% CI 28.0–51.4), 58.7% (95% CI 46.2–71.6), 65.5% (95% CI 52.6–78.0), and 65.5% (95% CI 52.6–78.0) in the overweight patients, respectively. No significant difference was observed between the two groups (HR 1.14, 95% CI 0.76–1.70, *p* = 0.5; Figure 4a). The IPTW analysis showed that the 1-, 2-, 3-, and 4-year distant recurrence rates were 33.7% (95% CI 25.2–44.1), 51.8% (95% CI 41.3–63.2), 59.3% (95% CI 47.4–71.5), and 63.1% (95% CI 50.2–76.1) in the non-overweight patients and 36.5% (95% CI 25.8–49.8), 53.7% (95% CI 40.9–67.6), 63.3% (95% CI 49.9–76.7), and 71.3% (95% CI 53.5–87.0) in the overweight patients, respectively. The difference was not statistically significant (HR 1.13, 95% CI 0.95–1.35, *p* = 0.2; Figure 4b).

Thirty patients (15.2%) died during the follow-up period. The 1-, 2-, 3-, and 4-year survival rates were 95.6% (95% CI 89.7–98.1), 90.8% (95% CI 82.7–95.2), 83.8% (95% CI 72.9–90.5), and 52.4% (95% CI 33.9–67.9) in the non-overweight patients and 98.4% (95% CI 89.4–99.8), 94.6% (95% CI 84.2–98.3), 87.7% (95% CI 74.3–94.4), and 87.7% (95% CI 74.3–94.4) in the overweight patients, respectively. No patient in either the overweight or non-overweight group reached the median OS. The survival curve was better in the overweight patients than in the non-overweight patients (HR 0.37, 95% CI 0.16–0.87, *p* = 0.02; Figure 5a). In the weighted cohort, the 1-, 2-, 3-, and 4-year survival rates were 95.2% (95% CI 91.2–99.4), 90.1% (95% CI 83.9–96.7), 84.8% (95% CI 76.6–93.7), and 54.3% (95% CI 38.9–75.8) in the non-overweight patients and 98.3% (95% CI 94.9–100), 93.9% (95% CI 87.3–100), 85.7% (95% CI 75.4–97.3), and 85.7% (95% CI 75.4–97.3) in the overweight patients, respectively. The OS rate was higher in the overweight patients than in the non-overweight patients (HR 0.36, 95% CI 0.25–0.51, *p* < 0.001; Figure 5b).

## 4. Discussion

The major finding of the present study was that the TSR in the first session was significantly lower in the overweight patients than in the non-overweight patients. We performed additional ablation therapy for the residual tumors, showing that the TSR at the final session was comparable between the overweight and non-overweight patients. The survival curve was better in the overweight patients than in the non-overweight patients, although local tumor progression and distant recurrence rates were not significantly different. In short, while the overweight patients were not likely to achieve a high TSR in the first session, especially in cases with tumor spread to both lobes and multiple nodules, they were expected to achieve better survival compared to non-overweight patients. Despite the potential adverse impact of being overweight on public health problems, the present findings showed the relationship between being overweight and improved survival. The negative aspects of being overweight might remain as minor technical issues in HCC patients receiving ablation therapy.

Ohki et al. reported that overweight patients receiving RFA required more treatment sessions for complete ablation compared to non-overweight patients [18], which is consistent with the present results. In our further subgroup analyses, overweight patients with tumors that spread to both lobes and those with multiple nodules were unlikely to achieve a high TSR in the first session. This was probably because subcutaneous fat reduced the resolution of ultrasonography and hindered clear visualization of targeted tumors. We compared the predictive ability of SATI and VATI, indicating that SAT might have a greater effect on the TSR in the first session than VAT. The difference between SAT and VAT was not statistically significant, which might be due to the lower statistical power of the present study.

Ohki et al. also reported that local tumor progression and overall recurrence rates were comparable between overweight and non-overweight patients [18]. Through multivariate analysis, they showed that being overweight was not a predictive factor for overall recurrence [18]. Guo et al. compared the clinical outcomes of overweight and non-overweight patients receiving curative resection and reported that disease-free survival was not significantly different between the two groups [19]. These previous reports agree with the results of the present study. However, we observed better OS in the overweight patients than in the non-overweight patients, which is inconsistent with the previous study by Ohki et al. [18]. This discrepancy may be due to differences in patient characteristics. In the present cohort, liver function appeared to be better (Child–Pugh class A, *n* = 174 (87.9%) vs. *n* = 511 (68.7%)) and the number of patients with solitary HCC seemed to be higher (*n* = 123 (62.1%) vs. *n* = 422 (56.8%)). Another reason might be the development of antiviral therapy for HBV [20,21] and HCV [22], which alleviates chronic liver inflammation and improves liver function. In addition, recent advances in systemic therapy, including sorafenib [23], lenvatinib [24], and atezolizumab plus bevacizumab [25], might also affect the present results because some patients who progressed to an advanced stage after curative treatment were indicated for systemic therapy. It is somewhat paradoxical that obesity could be a favorable prognostic factor, yet it is associated with an increased risk of HCC [26,27,28]. This phenomenon is termed the “obesity paradox”, which is also observed in coronary heart disease [29] and end-stage renal disease [30]. Regarding non-viral-related HCC, the lowest point on the cubic spline hazard according to BMI was approximately 26 kg/m^2^, as reported in a nationwide survey in Japan [11]. In the present study, patients with sarcopenia might have been included in the non-overweight patients, which could adversely affect survival. Further studies are warranted to investigate the relationship between BMI and OS in patients with HCC.

Currently, CEUS is one of the most common detection techniques for HCC [31]. However, the percentage of the overweight patients using CEUS was lower than that of the non-overweight patients in the present study. This is because we did not only use CEUS in patients with a tumor located deep from the liver surface, but also in those with a tumor not visualized due to subcutaneous fat and liver steatosis.

The present study has some limitations. First, this study was conducted retrospectively at a single center. Second, the number of cases was small, and the observation period was relatively short. A further study with a larger number of cases and adequate observation will be needed to confirm the present results. Third, we did not assess oxidative stress which is associated with being overweight because of the retrospective study design. Fourth, we did not evaluate the patients’ daily diets. For example, the fish oil high-fat diet (HFD) is not associated with tumor development while the cocoa butter HFD is [32]. Similarly, we also did not assess a lifestyle intervention of dietary weight loss counseling and moderate exercise. Whether or not purposeful weight loss after ablation therapy improves survival remains uncertain. Future research will be needed to evaluate the value of purposeful weight loss. Another limitation is that careful interpretation is required for adapting the present results for patients with BMI >30 kg/m^2^ because there were only 12 (6.1%) patients with BMI >30 kg/m^2^.

## 5. Conclusions

Despite the potential adverse impact of being overweight on public health problems, the present findings showed the relationship between being overweight and improved survival. The negative aspects of being overweight might remain as minor technical issues in HCC patients receiving ablation therapy.

## Figures and Tables

**Figure 1 cancers-15-01289-f001:**
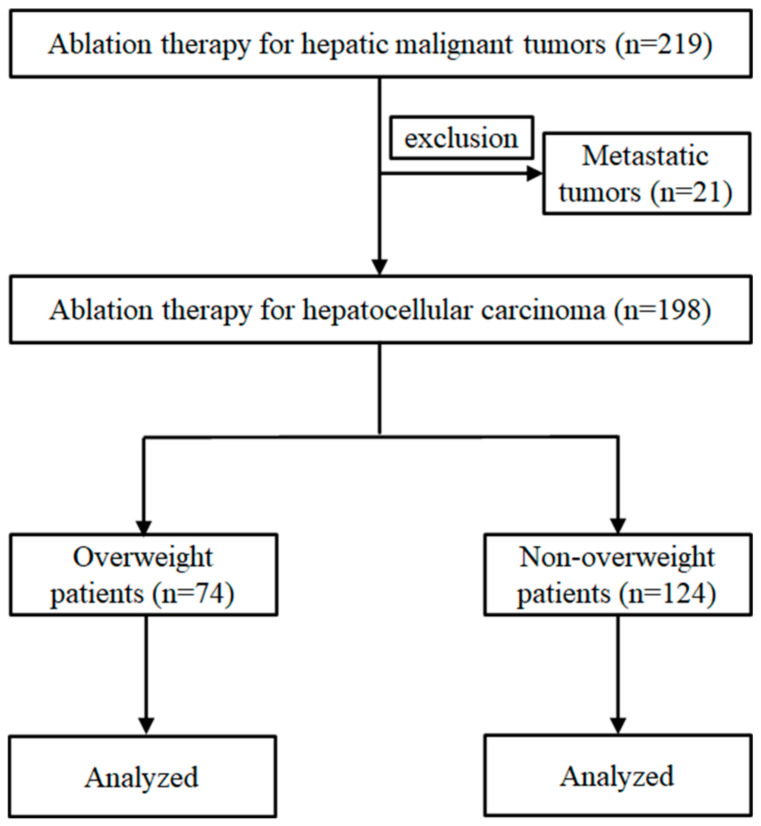
Patient selection flow chart.

**Figure 2 cancers-15-01289-f002:**
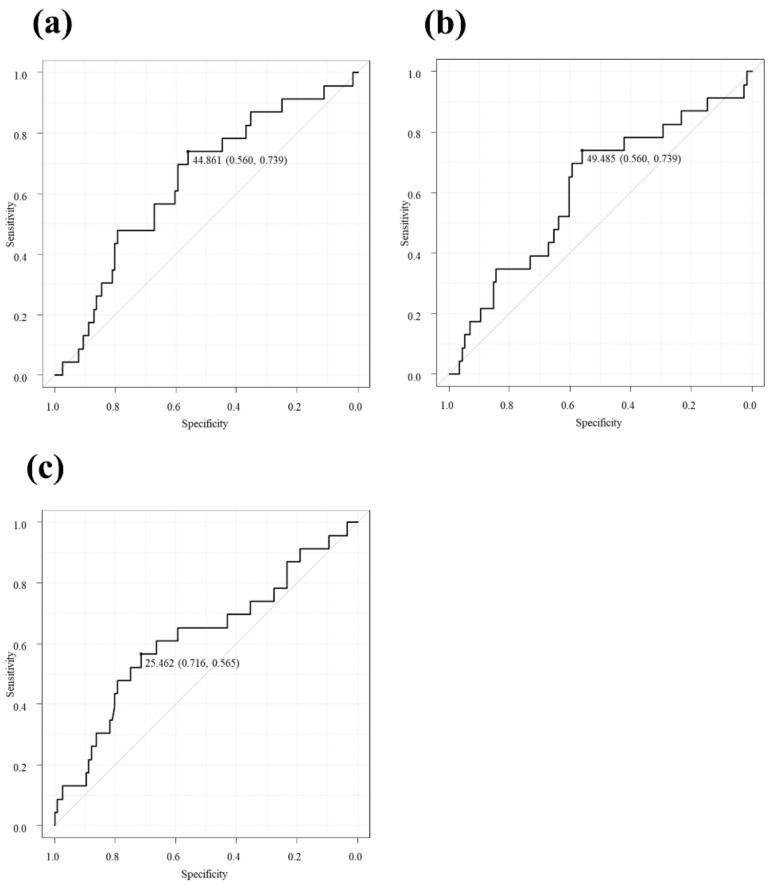
(**a**) Diagnostic performance of subcutaneous adipose tissue index (SATI) for the technical success rate at the first session. SATI had an AUROC of 0.64 (95% CI 0.52–0.76), with a specificity of 56.0% and a sensitivity of 73.9%. (**b**) Diagnostic performance of visceral adipose tissue index (VATI) for the technical success rate at the first session. VATI showed an AUROC of 0.61 (95% CI 0.48–0.74), with a specificity of 56.0% and a sensitivity of 73.9%. (**c**) Diagnostic performance of body mass index (BMI) for the technical success rate at the first session. BMI showed an AUROC of 0.62 (95% CI 0.48–0.76), with a specificity of 71.6% and a sensitivity of 56.5%. No statistically significant difference was observed between the three groups (SATI vs. VATI, *p* = 0.6; SATI vs. BMI, *p* = 0.6; VATI vs. BMI, *p* = 0.9). AUROC, area under the receiver operating characteristic curve.

**Figure 3 cancers-15-01289-f003:**
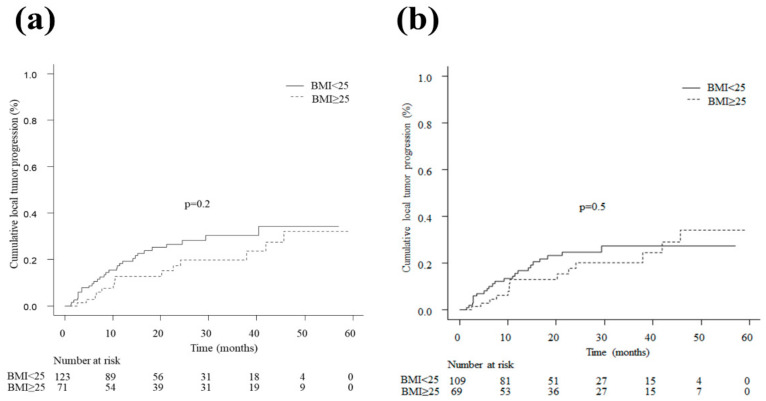
(**a**) Local tumor progression in patients with BMI < 25 kg/m^2^ and with BMI ≥ 25 kg/m^2^ in the crude cohort. The two groups showed no significant difference (HR 0.70, 95% CI 0.38–1.31, *p* = 0.2). (**b**) Local tumor progression in patients with BMI < 25 kg/m^2^ and with BMI ≥ 25 kg/m^2^ in the weighted cohort. No significant difference was observed between the two groups (HR 0.90, 95% CI 0.69–1.20, *p* = 0.50). BMI, body mass index; CI, confidence interval; HR, hazard ratio.

**Figure 4 cancers-15-01289-f004:**
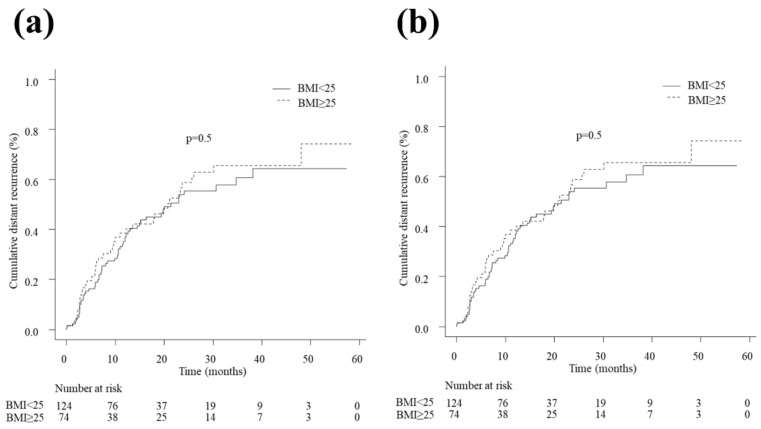
(**a**) Distant recurrence in patients with BMI < 25 kg/m^2^ and with BMI ≥ 25 kg/m^2^ in the crude cohort. No significant difference was observed between the two groups (HR 1.14, 95% CI 0.76–1.70, *p* = 0.5). (**b**) Distant recurrence in patients with BMI < 25 kg/m^2^ and with BMI ≥ 25 kg/m^2^ in the weighted cohort. The difference between the two groups was not significant (hazard ratio (HR) 1.13, 95% CI 0.95–1.35, *p* = 0.2). BMI, body mass index; CI, confidence interval; HR, hazard ratio.

**Figure 5 cancers-15-01289-f005:**
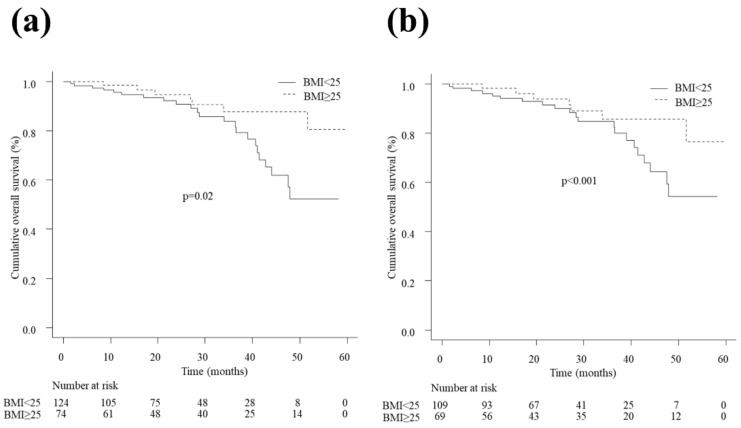
(**a**) Overall survival in patients with BMI < 25 kg/m^2^ and with BMI ≥ 25 kg/m^2^ in the crude cohort. The survival curve was better in overweight patients than in non-overweight patients (HR 0.37, 95% CI 0.16–0.87, *p* = 0.02). (**b**) Overall survival in patients with BMI < 25 kg/m^2^ and with BMI ≥ 25 kg/m^2^ in the weighted cohort. The OS rate was higher in overweight patients than in non-overweight patients (HR 0.36, 95% CI 0.25–0.51, *p* < 0.001). BMI, body mass index; CI, confidence interval; HR, hazard ratio; OS, overall survival.

**Table 1 cancers-15-01289-t001:** Patient characteristics.

	Overall Patients (*n* = 198)	Overweight Patients (*n* = 74)	Non-Overweight Patients (*n* = 124)	*p* Value
Age (years)	74.0 (69.0, 79.0)	73.5 (69.0–79.0)	74.0 (69.8–79.0)	0.6
Male, *n* (%)	142 (71.7)	55 (74.3)	87 (70.2)	0.6
Body mass index (kg/m^2^)	23.4 (22.0, 26.5)	27.2 (25.9–28.7)	22.5 (20.7–23.7)	<0.001
Underlying liver diseases, *n* (%)				0.02
HBV	11 (5.6)	1 (1.4)	10 (8.1)	
HCV	110 (55.6)	37 (50.0)	73 (58.9)	
Alcohol	42 (21.2)	16 (21.6)	26 (21.0)	
Others	35 (17.7)	20 (27.0)	15 (12.1)	
Viral-related disease, *n* (%)	121 (61.1)	38 (51.4)	83 (66.9)	0.04
Performance status, *n* (%)				0.7
0	160 (80.8)	60 (81.1)	100 (80.6)	
1	33 (16.7)	13 (17.6)	20 (16.1)	
2	5 (2.5)	1 (1.4)	4 (3.2)	
Hypertension, *n* (%)	125 (63.1)	47 (63.5)	78 (62.9)	1.0
Diabetes mellitus, *n* (%)	68 (34.3)	28 (37.8)	40 (32.3)	0.4
Dyslipidemia, *n* (%)	39 (19.7)	20 (27.0)	19 (15.3)	0.06
Treatment naïve, *n* (%)	84 (42.4)	28 (37.8)	56 (45.2)	0.4
Platelet count (×10^4^/μL)	13.9 (10.0–18.2)	12.7 (9.9–16.4)	14.0 (10.2–18.4)	0.2
Prothrombin time (%)	90.0 (77.8–97.5)	86.1 (75.8–95.1)	89.6 (80.8–97.7)	0.2
Total bilirubin (mg/dL)	0.9 (0.6–1.0)	0.9 (0.6–1.1)	0.8 (0.6–1.0)	0.1
Serum albumin (g/dL)	3.9 (3.6–4.2)	3.9 (3.6–4.1)	3.9 (3.6–4.2)	0.4
AST (U/L)	29 (22–39)	30 (22–40)	28 (22–38)	0.5
ALT (U/L)	22 (16–32)	24 (17–35)	21 (15–31)	0.09
Total lymphocyte count (/μL)	1310 (985–1730)	1400 (1075–1780)	1280 (905–1670)	0.08
Total cholesterol (mg/dL)	169 (143–191)	169 (256–187)	169 (142–193)	0.8
Child–Pugh Class, *n* (%)				0.5
A	174 (87.9)	65 (87.8)	109 (87.9)	
B	22 (11.1)	9 (12.2)	13 (10.5)	
C	2 (1.0)	0 (0.0)	2 (1.6)	
ALBI score	−2.57 (−2.84 to −2.23)	−2.52 (−2.78 to −2.20)	−2.60 (−2.85 to −2.25)	0.3
ALBI grade, *n* (%)				0.2
1	94 (47.5)	32 (43.2)	62 (50.0)	
2	101 (51.0)	42 (56.8)	59 (47.6)	
3	3 (1.5)	0 (0.0)	3 (2.4)	
Maximum tumor diameter (mm)	18 (12–25)	18 (13–23)	18 (12–25)	0.9
The number of tumors, *n* (%)				0.7
1	123 (62.1)	43 (58.1)	80 (64.5)	
2	42 (21.2)	17 (23.0)	25 (20.2)	
3	23 (11.6)	9 (12.2)	14 (11.3)	
≥4	10 (5.1)	5 (6.8)	5 (4.0)	
AFP ≥ 100 (ng/mL), *n* (%)	20 (10.2)	4 (5.4)	16 (13.1)	0.09
AFP-L3 ≥ 20 (%), *n* (%)	24 (12.2)	5 (6.8)	19 (15.6)	0.08
DCP ≥ 100 (mAU/mL), *n* (%)	27 (14.1)	9 (12.5)	18 (15.1)	0.7
Treatment modality, *n* (%)				0.2
Radiofrequency ablation	138 (69.7)	56 (75.7)	82 (66.1)	
Microwave ablation	60 (30.3)	18 (24.3)	42 (33.9)	

AFP, α-fetoprotein; AFP-L3, lens culinaris agglutinin–reactive fraction of α-fetoprotein; ALBI grade, albumin-bilirubin grade; ALBI score, albumin-bilirubin score; ALT, alanine aminotransferase; AST, aspartate aminotransferase; DCP, des-gamma-carboxy prothrombin; HBV, hepatitis B virus; HCV, hepatitis C virus.

**Table 2 cancers-15-01289-t002:** The short-term results of ablation therapy.

	Overweight Patients (*n* = 74)	Non-Overweight Patients (*n* = 124)	*p* Value
TSR (%)			
at the first session	78.4	90.3	0.03
at the final session	95.9	99.2	0.3
Additional ablation, *n* (%)	15 (20.3)	11 (8.9)	0.03
CEUS, *n* (%)	23 (31.1)	61 (49.2)	0.02
Fusion imaging, *n* (%)	70 (94.6)	110 (88.7)	0.2
The number of insertions	3 (2–4)	3 (2–4)	0.3
Total ablation time (min)	11 (6.3–15.9)	10 (6.0–15.0)	0.5
Procedure time (min)	61.5 (46.3–87.5)	67.0 (48.5–88.0)	0.7
Dose of midazolam (mg)	6 (5–8)	6 (4–7)	0.1
Dose of pentazocine (mg)	30 (15–30)	30 (15–30)	0.3
Major complication, *n* (%)	1 (1.4)	2 (1.6)	1.0

CEUS, contrast-enhanced ultrasonography; TSR, technical success rate.

**Table 3 cancers-15-01289-t003:** Subgroup analyses of tumor factors affecting the TSR at the first session.

	Overweight Patients (*n* = 74)	Non-Overweight Patients (*n* = 124)	*p* Value
Overall patients	78.4 (58/74)	90.3 (112/124)	0.03
Maximum tumor diameter (mm)			
<20	88.4 (38/43)	95.6 (65/68)	0.3
≥20	64.5 (20/31)	83.9 (47/56)	0.06
Tumor spread *			
Left lobe	81.2 (13/16)	90.0 (18/20)	0.6
Right lobe	95.1 (39/41)	93.2 (82/88)	1.0
Both lobes	31.2 (5/16)	75.0 (12/16)	0.03
The number of tumors			
Solitary	93.0 (40/43)	95.0 (76/80)	0.7
Multiple	58.1 (18/31)	81.8 (36/44)	0.04
Left or right lobe	86.7 (13/15)	85.7 (24/28)	1.0
Both lobes	31.2 (5/16)	75.0 (12/16)	0.03

* One overweight patient had a tumor nodule in the caudate lobe.

## Data Availability

All data supporting the present study are available from the corresponding author upon reasonable request.

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
