# Peer review of "Relationship between Being Overweight and Clinical Outcomes of Ablation Therapy for Hepatocellular Carcinoma under Ultrasound Guidance: A Retrospective Analysis"

_cancers, 2023, doi:10.3390/cancers15041289_

Round 1

Reviewer 1 Report

The effect of obesity on the outcome of ablation therapy for patients with early-stage hepatocellular carcinoma (HCC).

This retrospective study included 198 patients with HCC who underwent radiofrequency ablation or microwave ablation at Gunma Saiseikai Maebashi Hospital between April 2017 and December 2021. The patients into two groups based on their body mass index (BMI): obese (BMI ≥ 25, n=74 [37.4%]) and non-obese (BMI < 25, n=124 [62.6%]). The technical success rates (TSR) in the first session were 78.4% and 90.3% in obese and non-obese patients, respectively, with significant differences (p=0.03). Additional ablation therapy for residual tumors was required in 15 (20.3%) obese and 11 (8.9%) non-obese patients (p=0.03), resulting in 95.9% and 99.2% TSR at the final session, respectively, without a significant difference (p=0.3). While local tumor progression and distant recurrence rates were not significantly different between the two groups, overall survival was better in obese patients than in non-obese patients (p<0.001). Although additional ablation therapy is required for obese patients, obesity could be a favorable prognostic factor, and physicians should not avoid ablation therapy solely because of obesity.

The retrospective study is well design and the result is credible. However, there still have some issues need to be clarified.

1.      Obese patients receiving RFA required more treatment sessions for complete ablation compared to non-obese patients. The diet or food factors play an important role in the procession

2.      Oxidative stress state should be evaluated in the daily diet(The positive correlation of antioxidant activity and prebiotic effect about oat phenolic compounds. )

3.       Obesity often accompanied with metabolic syndrome. Please supplement such as FABP4 or FABP5(Consumption of the fish oil high-fat diet uncouples obesity and mammary tumor growth through induction of reactive oxygen species in pro-tumor macrophages. Cancer Research, 2020, 80(12): 2564-2574.)

4.      The grammar issues need to check.

Author Response

Dear editor and reviewers

We would like to thank the editors and reviewers for the insightful comments, which have been very helpful in improving our manuscript ‘cancers-2245641’. We have been able to incorporate changes to reflect the suggestions provided by the reviewers.

Here is a point-by-point response to the reviewers’ comments and concerns.

Reviewer 1

This retrospective study included 198 patients with HCC who underwent radiofrequency ablation or microwave ablation at Gunma Saiseikai Maebashi Hospital between April 2017 and December 2021. The patients into two groups based on their body mass index (BMI): obese (BMI ≥ 25, n=74 [37.4%]) and non-obese (BMI < 25, n=124 [62.6%]). The technical success rates (TSR) in the first session were 78.4% and 90.3% in obese and non-obese patients, respectively, with significant differences (p=0.03). Additional ablation therapy for residual tumors was required in 15 (20.3%) obese and 11 (8.9%) non-obese patients (p=0.03), resulting in 95.9% and 99.2% TSR at the final session, respectively, without a significant difference (p=0.3). While local tumor progression and distant recurrence rates were not significantly different between the two groups, overall survival was better in obese patients than in non-obese patients (p<0.001). Although additional ablation therapy is required for obese patients, obesity could be a favorable prognostic factor, and physicians should not avoid ablation therapy solely because of obesity.

The retrospective study is well design and the result is credible. However, there still have some issues need to be clarified.

  1. Obese patients receiving RFA required more treatment sessions for complete ablation compared to non-obese patients. The diet or food factors play an important role in the procession.

Response

We thank you for your insightful comment.

We did not assess the diet and food factors in the present study because of retrospective study design. We discussed in the paragraph of limitation and added following sentence in line 370-375.

Fourth, we did not evaluate the patients’ daily diet. For example, the fish oil high fat diet (HFD) is not associated with tumor development while the cocoa butter HFD is [32]. In this connection, we also did not assess a lifestyle intervention of dietary weight loss counselling and moderate exercise. Whether or not purposeful weight loss after ablation therapy improves the survival remains uncertain. A future research will be needed to evaluate the value of purposeful weight loss.

  1. Oxidative stress state should be evaluated in the daily diet (The positive correlation of antioxidant activity and prebiotic effect about oat phenolic compounds. )

Response

We thank you for bringing this interesting point to our attention.

We fully understand the importance of oxidative stress. But, unfortunately, we did not assessed the oxidative stress.

We added the following sentences in line 368-369.

Third, we did not assess oxidative stress which are associated with overweight because of retrospective study design.

  1. Obesity often accompanied with metabolic syndrome. Please supplement such as FABP4 or FABP5(Consumption of the fish oil high-fat diet uncouples obesity and mammary tumor growth through induction of reactive oxygen species in pro-tumor macrophages. Cancer Research, 2020, 80(12): 2564-2574.)

Response

We thank you for your comment.

As reviewer 2 suggested, we added the metabolic comorbidity diseases such as hypertension, diabetes mellitus, and dyslipidemia in Table 1. We cited the manuscript as you mentioned, and discussed it in line 370-372.

For example, the fish oil high fat diet (HFD) is not associated with tumor development while the cocoa butter HFD is [32].

  1. The grammar issues need to check.

Response

Thank you for your comments.

We fully check the grammar issues with careful caution.

Reviewer 2 Report

Thank you for submitting "Relationship between obesity and clinical outcomes of ablation therapy for hepatocellular carcinoma under ultrasound guidance: A retrospective analysis" to Cancers.

In general, the research is interesting, but it is important to point out that it is not a question of separating an obese or non-obese population, since in both groups people would have different nutritional needs due to their nutritional status and this was not considered in this research.

The fact that there were no numbers on the lines made it difficult to point out specific revision locations.

Simple Summary

- "two groups based on body mass index" please enter information about the groups. There is no need to make the reader curious.

- "Our findings show that obesity could be a favorable prognostic factor" this statement was strange and it looks like researchers are favoring obesity which is a chronic disease considered a public health problem.

Abstract

- It is not necessary to indicate where each session begins. The abstract is expected to have this format.

Material and methods

- Were these non-obese patients graded for BMI?

- Consider using inclusive naming throughout your text: https://www.obesityaction.org/action-through-advocacy/weight-bias/people-first-language/

Discussion

- "and physicians should not avoid ablation therapy solely because of obesity." It seems to me that it is still too early to make this generalization.

- "The present study has some limitations." include limitations according to nutritional status such as lack of subgroups within the BMI, indication of other comorbidities, monitoring of patients' diet, etc.

Author Response

Dear editor and reviewers

We would like to thank the editors and reviewers for the insightful comments, which have been very helpful in improving our manuscript ‘cancers-2245641’. We have been able to incorporate changes to reflect the suggestions provided by the reviewers.

Here is a point-by-point response to the reviewers’ comments and concerns.

Reviewer 2

Thank you for submitting "Relationship between obesity and clinical outcomes of ablation therapy for hepatocellular carcinoma under ultrasound guidance: A retrospective analysis" to Cancers.

In general, the research is interesting, but it is important to point out that it is not a question of separating an obese or non-obese population, since in both groups people would have different nutritional needs due to their nutritional status and this was not considered in this research.

The fact that there were no numbers on the lines made it difficult to point out specific revision locations.

Response

We apologized for your inconvenience. We made numbers on the lines.

Simple Summary

- "two groups based on body mass index" please enter information about the groups. There is no need to make the reader curious.

Response

We thank you for your comment.

We added group name as the overweight and non-overweight patients. We changed sentences in line 17-19 as followed.

participants were divided into two groups based on body mass index (BMI): overweight patients (BMI ≥ 25) and non-overweight patients (BMI < 25).

- "Our findings show that obesity could be a favorable prognostic factor" this statement was strange and it looks like researchers are favoring obesity which is a chronic disease considered a public health problem.

Response

We thank you for your insightful comment. We agreed to your comment.

We deleted this statement and changed as follow in line 25-28, 43-46, 318-321, and 379-382.

Despite the potential adverse impact of overweight on public health problem, present findings showed the relationship between overweight and improved survival. The negative aspects of overweight might remain a minor technical issue in HCC patients receiving ablation therapy.

Abstract

- It is not necessary to indicate where each session begins. The abstract is expected to have this format.

Response

We thank you for your comment.

We deleted the ‘Aims’, ‘Methods’, ‘Results’, and ‘Conclusions’.

Material and methods

- Were these non-obese patients graded for BMI?

Response

Thank you for your comment.

We added following sentence in line 173-175.

Among non-overweight patients, 21 (10.6%) patients were underweight (<20 kg/m2) and 103 (52.2%) patients normal-weight (20–24.9kg/m2).

In addition, we changed the definition of overweight from the Japan Society for the Study of Obesity to World Health Organization. We changed the sentence in line 119-120 as follow.

We defined patients with BMI ≥ 25kg/m2 as overweight patients, and those with BMI < 25kg/m2 as non-overweight based on the criteria of World Health Organization [15].

- Consider using inclusive naming throughout your text: https://www.obesityaction.org/action-through-advocacy/weight-bias/people-first-language/

Response

Thank you for your insightful comment.

We replaced word ‘obesity’ to ‘overweight’ throughout our text including title, graphical abstract, and Figure 1.

Discussion

- "and physicians should not avoid ablation therapy solely because of obesity." It seems to me that it is still too early to make this generalization.

Response

Thank you for your meaningful comment.

We deleted this sentence. We added the following sentence in line 27-28, 45-46, 318-321, and 381-382.

The negative aspects of overweight might remain a minor technical issue in HCC patients receiving ablation therapy.

Moreover, we added following sentence in line 375-377.

Another limitation is that careful interpretation is required for adapting the present results for patients with BMI >30kg/m2 because there were only 12 (6.1%) patients with BMI >30kg/m2.

- "The present study has some limitations." include limitations according to nutritional status such as lack of subgroups within the BMI, indication of other comorbidities, monitoring of patients' diet, etc.

Response

Thank you for your comment.

As reviewer 1 also suggested, we added the metabolic comorbidity diseases such as hypertension, diabetes mellitus, and dyslipidemia in Table 1. We also described the value of total lymphocyte count and total cholesterol as nutritional status. There were no significant differences between the two groups.

we did not assess a lifestyle intervention of dietary weight loss counselling and moderate exercise. Whether or not purposeful weight loss after ablation therapy improves the survival remains uncertain.

As reviewer 1 suggested, we cited the reference [32] and added following sentences in line 368-374.

Fourth, we did not evaluate the patients’ daily diet. For example, the fish oil high fat diet (HFD) is not associated with tumor development while the cocoa butter HFD is [32]. In this connection, we also did not assess a lifestyle intervention of dietary weight loss counselling and moderate exercise. Whether or not purposeful weight loss after ablation therapy improves the survival remains uncertain. A future research will be needed to evaluate the value of purposeful weight loss.

Round 2

Reviewer 1 Report

The author has responsed the reviewer's comment. It can be accepted in current status.